# Distribution and treatment needs of soil-transmitted helminthiasis in Bangladesh: A Bayesian geostatistical analysis of 2017-2020 national survey data

**Daniel J. F. Gerber[1,2], Sanjaya Dhakal[3], Md. Nazmul Islam[4], Abdullah Al Kawsar[4], Md. Abul Khair[4], Md. Mujibur Rahman[4], Md. Jahirul Karim[4], Md. Shafiqur Rahman[4], M. M. Aktaruzzaman[4], Cara Tupps[3], Mariana Stephens[3], Paul M. Emerson[3], Jürg Utzinger[1,2], Penelope Vounatsou[1,2] ***

**1** Swiss Tropical and Public Health Institute, Allschwil, Switzerland, **2** University of Basel, Basel, Switzerland, **3** Children Without Worms, Task Force for Global Health, Decatur, Georgia, United States of America, **4** Communicable Disease Control Unit, Directorate General of Health Services, Ministry of Health & Family Welfare, Dhaka, Bangladesh

* penelope.vounatsou@swisstph.ch

## Abstract

### Background

In Bangladesh, preventive chemotherapy targeting soil-transmitted helminth (STH) infections in school-age children has been implemented since 2008. To evaluate the success of this strategy, surveys were conducted between 2017 and 2020 in 10 out of 64 districts. We estimate the geographic distribution of STH infections by species at high spatial resolution, identify risk factors, and estimate treatment needs at different population subgroups.

### Methodology

Bayesian geostatistical models were fitted to prevalence data of each STH species. Climatic, environmental, and socioeconomic predictors were extracted from satellite images, open-access, model-based databases, and demographic household surveys, and used to predict the prevalence of infection over a gridded surface at 1 x 1 km spatial resolution across the country, via Bayesian kriging. These estimates were combined with gridded population data to estimate the number of required treatments for different risk groups.

### Principal findings

The population-adjusted prevalence of *Ascaris lumbricoides*, *Trichuris trichiura*, and hookworm across all ages is estimated at 9.9% (95% Bayesian credible interval: 8.0-13.0%), 4.3% (3.0-7.3%), and 0.6% (0.4-0.9%), respectively. There were 24 out of 64 districts with an estimated population-adjusted STH infection prevalence above 20%. The proportion of households with improved sanitation showed a statistically important, protective association for both, *A. lumbricoides* and *T. trichiura* prevalence. Precipitation in the driest month of the year was negatively associated with *A. lumbricoides* prevalence. High organic carbon

**Data Availability Statement:** All relevant data are within the previously published manuscript of some of coauthors (Dhakal et al. 2020, https://doi.org/10.1371/journal.pntd.0008597).

**Funding:** The study was funded by Children without Worms (Project ID: CWW-JNJ-S1). The funders had no role in study design, data collection and analysis, decision to publish, or preparation of the manuscript.

**Competing interests:** The authors have declared that no competing interests exist.

concentration in the soil's fine earth fraction was related to a high hookworm prevalence. Furthermore, we estimated that 30.5 (27.2; 36.0) million dosages of anthelmintic treatments for school-age children were required per year in Bangladesh.

## Conclusions/significance

For each of the STH species, the prevalence was reduced by at least 80% since treatment was scaled up more than a decade ago. The current number of deworming dosages could be reduced by up to 61% if the treatment strategy was adapted to the local prevalence.

## Author summary

Bangladesh was in 2003 one of the world's countries that was most affected by soil-transmitted helminthiasis, an infection of parasitic intestinal worms, with prevalence estimates as high as 55%. Following recommendations put forth by the World Health Organization (WHO), Bangladesh initiated a national deworming program based on school mass drug administration at six-month intervals, which started in 2008. Data collected between 2017 and 2020 showed that this program has lead to a striking decrease in prevalence, which was now between 10.5% and 1% depending on the species. Entering a close to elimination setting, the disease's control strategy can be refined according to local endemicity to reduce overtreatment. To inform such a shift, we predict the prevalence across Bangladesh at high spatial resolution and highlight zones of different endemicity. We also show the potential reduction of overtreatment and financial resources that could be achieved by treating according to WHO recommendations at the predicted endemicity levels.

## Introduction

The number of soil-transmitted helminth (STH) infection worldwide was estimated at around 900 million people in 2017 [1]. If untreated, STH infections can lead to intestinal obstruction, abdominal pain or bleeding, anemia, and malnourishment [2, 3]. This is particularly harmful for the physical and mental development of children and the health of pregnant women [4, 5]. In 2015, the three STH species, i.e. roundworm (*Ascaris lumbricoides*), whipworm (*Trichuris trichiura*), and hookworm (*Ancylostoma duodenale* and *Necator americanus*) caused a loss of 1.1, 0.5, and 1.8 million disability-adjusted life years (DALYs), respectively [6]. Affecting predominantly poor and marginalized communities in low- and middle-income countries, STH infection are at the same time a cause and effect of poverty. To break this vicious cycle, the World Health Organization (WHO) has included STH in the program for eliminating neglected tropical diseases (NTDs) in 2001 and suggested the use of preventive chemotherapy as the main intervention to achieve this goal [7].

In line with WHO recommendations, Bangladesh adopted the goal of eliminating STH infection as a public health problem and initiated a national deworming program based on school mass drug administration (MDA) at six-month intervals at elementary schools (age-groups 6–12 years), which started in 2008 [8]. Ongoing, this deworming program had reached a national treatment coverage increasing from 86.4% to 98.3% in the four year period 2015–2019 and has distributed a total of 78.2 million tablets in 2019 alone [9]. Until 2030, the WHO targets to reduce the number of needed tablets in the frame of MDA to preschool-age children (PSAC, younger than 5 years old) and school-age children (SAC, between 5 and 14 years old)

and to establish an efficient STH control program in adolescent, pregnant and lactating women of reproductive age (WRA, between 15 and 49 years old women) [10]. To reach these targets, WHO recommends to reassess the prevalence and to update the MDA treatment frequency accordingly. To this end and to evaluate the differences in prevalence for the aforementioned risk groups, the Bangladesh Ministry of Health & Family Welfare (MOHFW), together with Children Without Worms (CWW), developed the *Integrated Community-based Survey for Program Monitoring* (ICSPM). This survey was implemented in 10 districts between 2017 and 2020 and results were reported for 10 districts [11, 12]. The raw survey data indicated a sharp drop in the STH infection prevalence over the past two decades. Indeed, in 2003, the prevalence of *A. lumbricoides*, *T. trichiura*, and hookworm in the population of Bangladesh was 55.0%, 46.0%, and 35.0%, respectively [13]. The data of 2017–2020 revealed that the prevalence for the aforementioned species was reduced to 10.5%, 4.4%, and less than 1%, respectively [12]. The district-level aggregated data suggested that the geographic distribution of the prevalence was not homogeneous across the country, despite a uniform MDA distribution. Climatic and socioeconomic factors might explain the observed geographic disparities. Moreover, it is unclear how the disease distribution evolved outside the surveyed districts.

In view of progress made against STH infection, the question arises whether continued MDA is sensible for the entire country or whether a more focal approach would be more efficient and cost-effective. Bayesian geostatistical methodology (BGM) allows estimation of disease focality, as it takes into account the relation between the disease prevalence and the potential environmental and socioeconomic exposures on one hand and the geographic correlation in the data, that is nearby communities exhibit more similar STH infection prevalence than more distant ones, on the other hand. The model assesses risk factors, which drive the geographic distribution of the disease and predicts the prevalence at unobserved locations. Bayesian geostatistics has been successfully used to analyze the prevalence of STH infection in different parts of the world, including sub-Saharan Africa [14], South America [15]; as well as to analyze the situation in individual countries: Bolivia [16], Brazil [17], Cambodia [18], Côte d'Ivoire [19], Kenya [20], and the People's Republic of China [21], to highlight a few.

Using BGM, we analyzed the survey data from the ICSPM of 2017–2020 and provide scientific information on the current state of prevalence of STH infection in Bangladesh. Specifically, we (i) identify environmental and socioeconomic risk factors that are related to the geographic distribution of species-specific STH prevalence; (ii) predict the disease prevalence by species at high spatial resolution and detect high prevalence areas; (iii) analyze STH species-specific prevalence differences between the risk groups PSAC, SAC, adults (15 years old and above), and WRA; and (iv) estimate the number of tablets needed for MDA according to these estimated prevalence levels and selected implementation strategies.

## Methods

### Ethics statement

We report a secondary geostatistical analysis of STH survey data in Bangladesh. Participation in the original survey was voluntary and participants provided verbal consent before the main survey. Ethical clearance for the survey was obtained through the Bangladesh Medical Research Council (BMRC), who reviewed and approved the survey protocol.

### Data

The STH infection data were obtained from 10 community-based surveys, which were conducted in 10 districts, each taking 10 days in the time period between 2017 and 2020. Stool samples were subjected to duplicate Kato-Katz thick smears [22]. Details on the survey design

are provided elsewhere [12]. In this study, we defined the infection status of an individual according to the maximum egg count of the duplicate Kato-Katz thick smears. The individual infection status was aggregated at the community level to estimate the prevalence in a community. The unit of analysis was this community prevalence, which was georeferenced according to the centroid of the surveyed households within a community.

The survey data were combined with climatic, demographic, environmental, and socioeconomic variables, which were considered as predictors in our analysis. These variables were extracted from various sources, including satellites, model-based gridded surfaces, and Demographic Health Surveys (DHS), and used to derive 47 predictors. Land surface temperature (LST) was obtained from Moderate-resolution Imaging (MODIS) satellite as 8-day averages during 2015–2020 at 1 x 1 km spatial resolution. During the same period, 5-day averages of precipitation was extracted from the Climate Hazards Group InfraRed Precipitation with Station (CHIRPS) database at 5.4 x 5.4 km resolution. The LST and CHIRPS data were used to calculate long-term monthly averages over the period 2015–2020. The monthly estimates were further used to derive 19 bioclimatic variables (defined on www.worldclim.org), which capture temperature, precipitation, and their variation at different seasons of the year. Land cover (LC) data for 2019 at 100 x 100 m resolution was extracted from Copernicus Land Monitoring Service. The original 23 categories were regrouped into 5 categories (i.e., cropland, forest, shrubs/herbaceous vegetation, urban/barren, and permanent water) and converted into four predictors, indicating the surface covered within each district by the corresponding category. Due to collinearity, the categories cropland and shrubs/herbaceous vegetation were not included in the analysis. The DHS data of 2014 and 2017/2018 were used to calculate water, sanitation, and hygiene (WASH) indicators, i.e., proportion of households (i) with improved sanitation facilities; (ii) practicing open defecation; (iii) with improved drinking facilities; and (iv) with handwashing facility at the DHS survey locations. These proxies were aggregated at district level. Other predictors included agro-ecological zones, elevation, proxies of humidity (i.e., normalized difference vegetation index [NDVI] and enhanced vegetation index [EVI]), soil composition, and socioeconomic disparities measured by a household asset index calculated from DHS data and categorized in poverty quantiles. The aforementioned predictors were extracted or derived at the community locations. The LC and WASH indicators were linked to the survey data according to the district they belonged to. Finally, all calculations involving population relied on Data for Good at Meta (previously Facebook) population density data for 2020 at 100 x 100 m resolution [23]. These population density data are generated by combining census data with the number of buildings found on high resolution satellite images. An overview of the variables, their source, and spatial and temporal resolution and maps of selected variables are given in S1 Appendix.

## Statistical analysis

To address collinearity, we excluded covariates with a variance inflation factor above 10 [24]. For modeling the prevalence, species-specific Bayesian geostatistical logistic regression models with spatially structured and independent random effects were developed [25]. The unit of analysis was the community level prevalence. Details on the model specifications are provided in S2 Appendix.

To obtain parsimonious predictive models, variable selection was applied to identify the most important predictors of each STH species. In particular, Bayesian geostatistical bivariate regression models were fitted with one covariate at a time and the statistically important covariates (i.e., 95% Bayesian credible interval [BCI] did not include zero) were retained in the final multivariable model.

**Table 1. Mass drug administration guidelines for countries with more than 5–6 years preventive chemotherapy.**

| Area | Prevalence | MDA frequency |
|---|---|---|
| Very low endemic | $p < 2\%$ | Never |
| Low endemic | $2\% \leq p < 10\%$ | 1x every two years |
| Moderate endemic | $10\% \leq p < 20\%$ | 1x every year |
| High endemic | $20\% \leq p < 50\%$ | 2x every year |
| Very high endemic | $p \geq 50\%$ | 3x every year |

Model validation was performed by fitting each model to a randomly selected subset of 80% of the community locations (training set) and predicting the disease risk at the remaining ones (testing set). The sampling was repeated 40 times. The model's predictive ability was measured by the mean absolute error (MAE) and by assessing the proportion of testing locations with observed prevalence within the 95% BCI of the corresponding predictive posterior distribution. The prediction bias was measured by the mean error (ME) which was calculated by comparing the mean posterior predictive distribution at a given location with the observed prevalence and then averaging over all locations.

Bayesian kriging was used to predict the prevalence for each STH species over a gridded surface across Bangladesh. The posterior predictive distribution at each pixel (i.e., grid cell) was summarized by the median and graphically displayed to obtain a prevalence map at 1 x 1 km spatial resolution. The probability of being infected by any STH species was estimated by assuming independence of the infections risks of the three STH species.

The STH prevalence differences for PSAC, SAC, adults, and WRA were assessed by refitting the models including age as a categorical covariate (with categories PSAC, SAC, and adults), and a dichotomous variable to contrast WRA to the rest of the population, respectively. To estimate the number of infected people on the whole and for each risk group, we multiplied the prevalence with the population in each pixel and summed this over the administrative unit (district or entire country). We divided this number by the population living in this administrative unit to obtain the population-adjusted prevalence.

Treatment needs for 2020 were based on WHO's treatment guidelines for STH control programs summarized in Table 1 [10]. Concretely, the median predicted prevalence in each $1 \times 1$ km sized pixel was translated to the yearly treatment frequency according to Table 1 and subsequently multiplied with the pixel's population. The treatment needs were calculated in this way for the entire population and each risk group.

The statistical analysis was carried out using R software [26]. For estimation and prediction of the models, we used integrated nested Laplace approximation (INLA) implemented in the R-INLA package [27, 28].

## Results

The survey included 10,421 individuals in all age groups carried out in 296 villages in 10 out of 64 districts of Bangladesh. The surveyed districts were distributed rather evenly across the country with two notable exceptions: the divisions of Rangpur in the northwestern part and of Chittagong in the southeastern part of the country, which were under-represented. Overall, the observed prevalence of *A. lumbricoides*, *T. trichiura*, and hookworm was 10.2%, 4.2%, and 0.4% with 31.4%, 64.9%, and 87.2% of the locations featuring zero infection counts, respectively (Fig 1, left). The surveyed districts with the highest prevalence of *A. lumbricoides* were Sunamganj (29.2%), Sirajganj (25.1%), Bhola (22.1%), and Moulvibaz (11.2%). The prevalence

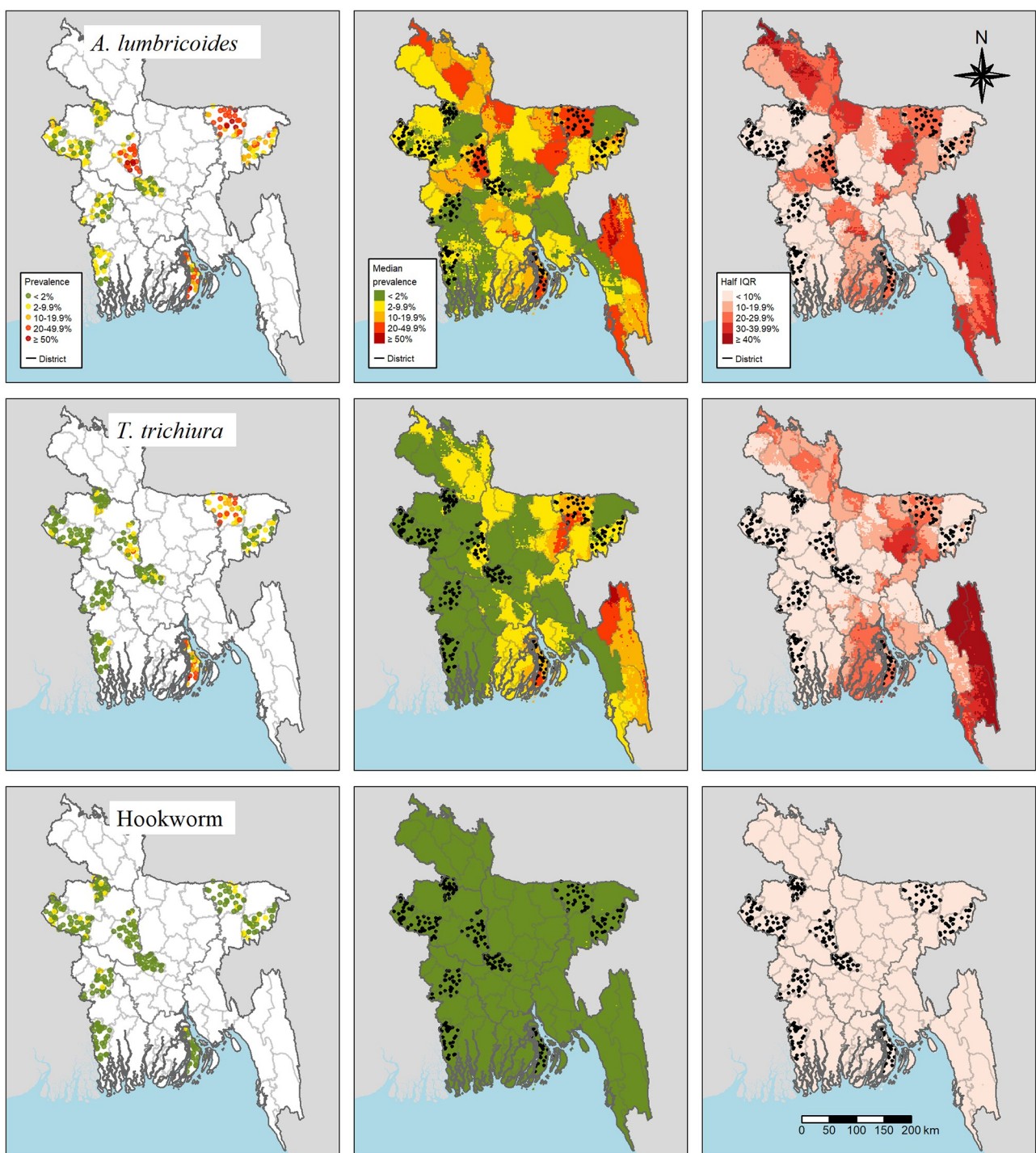

**Fig 1.** Maps of the observed prevalence (left), predicted prevalence (median of posterior predictive distribution; center), and prediction uncertainty (half interval length of 95% BCI interval; right) for the three STH species. These maps were created using R's tmap-package [29] and include administrative boundaries retrieved from gadm.org.

in the remaining districts was around 3% or lower. *T. trichiura*, was the most prevalent STH species in Bhola (18.5%) and Sunamganj (13.2%), while the prevalence in the other surveyed districts was 3% or lower. For hookworm, the two districts Manikganj and Satkhira recorded a 0% prevalence, while in the other districts a prevalence of 1% or lower was observed.

## Risk factors

The results from the variable selection process through bivariate geostatistical analysis in Fig 2 suggest that the factors with a statistically important protective effect (in statistical terms) on *A. lumbricoides* prevalence were the proportion of households per district with improved sanitation and hand washing facilities, the mean temperature of the warmest quarter, the temperature annual range, and the precipitation of the driest month of the year. However, the density of organic carbon in the soil was positively associated with the infection prevalence. The associations between climatic variables and prevalence of *T. trichiura* were similar to *A. lumbricoides*. In particular, a negative association was found with the annual mean temperature and the mean temperature of the warmest quarter. Furthermore, improved sanitation was related to reduced *T. trichiura* prevalence. A number of soil related parameters showed a statistically important association with hookworm prevalence. The association was negative for soils with high density of organic carbon, nitrogen, or cation exchange capacity and for the proportion of land surface covered with water, while it was positive for the proportion of organic carbon in the soil's fine earth fraction. Socioeconomic factors were not statistically important; however, the estimates of the poverty proxy had high uncertainty (i.e., large BCI).

Models including all variables from the variable selection process were fitted and their parameter estimates displayed in Table 2. The main predictors of *A. lumbricoides* prevalence were the proportion of households per district with improved sanitation (statistically

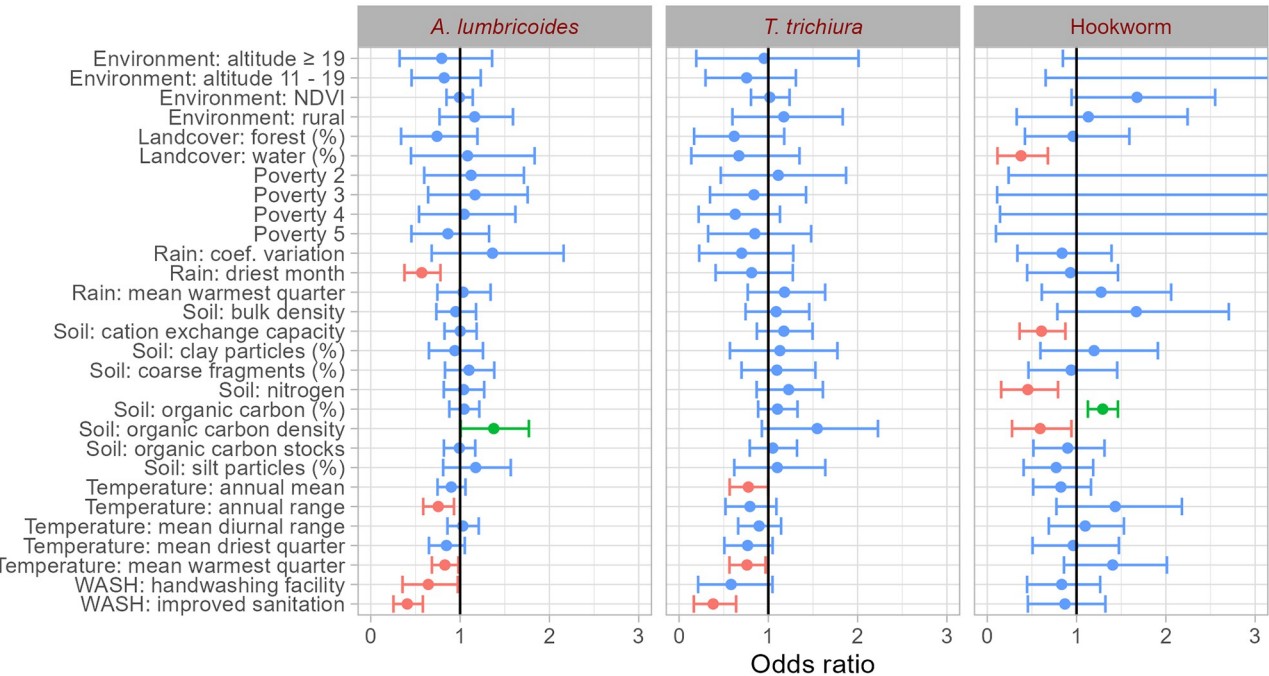

**Fig 2. Estimates (posterior mean and 95% BCI) of the effects of the risk factors obtained from STH species-specific, bivariate, geostatistical regression models used for variable selection purposes.**

**Table 2. Parameter estimates (posterior median and 95% BCI) of the final multivariable geostatistical models for each STH species.** The parameters for the regression coefficients are given as odds ratios.

| | *A. lumbricoides* | *T. trichiura* | Hookworm |
|---|---|---|---|
| Temperature: annual mean | | 0.87 (0.56; 1.20) | |
| Temperature: annual range | 0.90 (0.63; 1.19) | | |
| Temperature: mean warmest quarter | 0.95 (0.73; 1.19) | 0.87 (0.58; 1.19) | |
| Rain: driest month | 0.65 (0.45; 0.87) | | |
| Soil: cation exchange capacity | | | 1.10 (0.94; 1.26) |
| Soil: organic carbon density | 1.31 (0.98; 1.65) | | 1.01 (0.84; 1.19) |
| Soil: nitrogen | | | 0.96 (0.79; 1.14) |
| Soil: organic carbon (%) | | | 1.19 (1.00; 1.38) |
| WASH: improved sanitation | 0.24 (0.09; 0.43) | 0.40 (0.17; 0.68) | |
| WASH: handwashing facility | 2.54 (0.95; 4.39) | | |
| Land cover: water (%) | | | 1.11 (0.95; 1.29) |
| $\sigma^2_{\text{non-spatial}}$ | 0.27 (0.17; 0.42) | 0.38 (0.18; 0.69) | 0.39 (0.27; 0.56) |
| $\sigma^2_{\text{spatial}}$ | 0.82 (0.33; 1.72) | 2.24 (0.93; 4.61) | 0.14 (0.03; 0.43) |
| Range (km) | 128 (54; 264) | 149 (72; 279) | 424 (82; 1205) |
| Mixing probability $\theta$ | | | 0.87 (0.83; 0.90) |

important negative effect) and the amount of rainfall in the driest month of the year (negative effect). The proportion of households per district with improved sanitation was the only statistically important predictor of *T. trichiura* prevalence, indicating that the higher the proportion, the lower the prevalence. For hookworm, the higher the proportion of organic carbon in the soils fine earth fraction, the higher the prevalence.

The highest spatial variation was estimated for *T. trichiura*, followed by *A. lumbricoides*. The spatial variation was stronger than the non-spatial one for *T. trichiura* and *A. lumbricoides*; however, a rather larger non-spatial variation in the distribution of hookworm was found. The maximum distance at which spatial correlation was present was 149 km (95% BCI: 72; 279) and 128 km (54; 264) for *T. trichiura* and *A. lumbricoides*, respectively. There was considerable uncertainty in the estimation of this distance in the case of hookworm. Finally, the zero-inflated model for hookworm estimated a high mixing probability of 87% structural zeros.

Model validation showed that the MAE on average was 5.6%, 3.0%, and 0.7%, while for 61.4%, 66.5%, and 92.3% of the locations on average the observed prevalence fell within the 95%-BCI of the predictive posterior distribution for *A. lumbricoides*, *T. trichiura*, and hookworm, respectively. The average ME was 0.27%, 0.50%, and 0.00% for the respective species, suggesting that the models overestimate the prevalence slightly if at all.

### Geographic distribution

The geographic distribution of STH species-specific predicted prevalence and associated uncertainty based on a 1 x 1 km grid is visualized in Fig 1 (center and right). For *A. lumbricoides*, endemic regions with a prevalence above 20% were found in Bhola, Sirajganj, Sunamganj, as well as in the districts of Khagrachhati, Rangammati, and Bandarban, while all other districts featured a prevalence below 10% and in a substantial part even below 2%. For *T. trichiura*, endemic regions with a prevalence above 20% were found in Bhola, at the southern tip of the districts Netrakona and Sunamganj reaching into Kishoreganj, and in the northern part

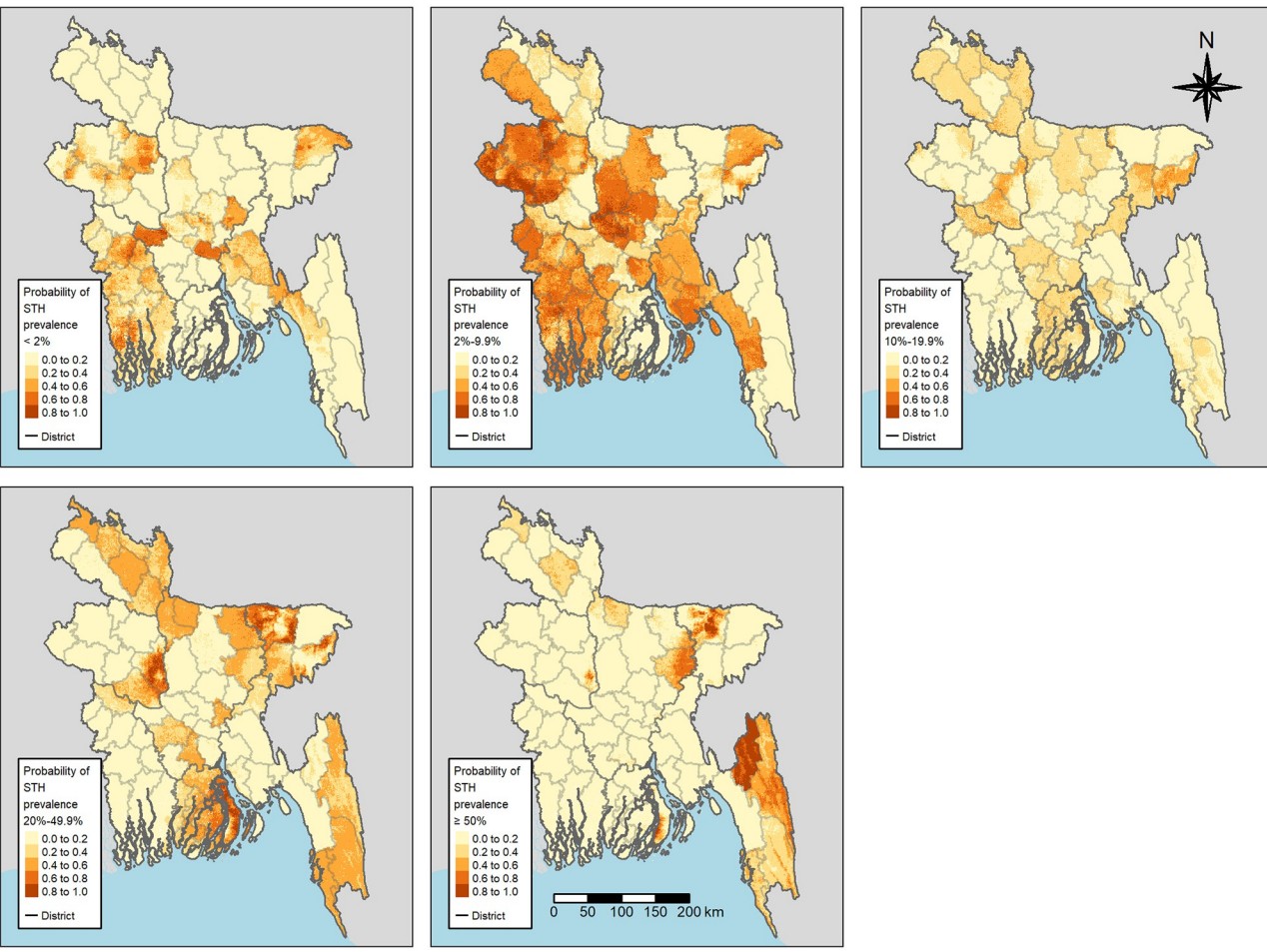

**Fig 3.** Posterior probabilities of very low (top left), low (top center), moderate (top right), high (bottom left), and very high (bottom right) prevalence of any STH infection in school-age children. These maps were created using R's tmap-package [29] and include administrative boundaries retrieved from gadm.org.

of the Chittagong region. All other districts revealed lower prevalence with about half of the country's surface showing a prevalence below 2%. Hookworm infection prevalence was under 2% across the entire country. Overall, the half length of the 95% BCI indicated that uncertainty of the prediction was high where the prevalence was high and even more so if the prevalence was high and survey locations were sparse, e.g., in the southeastern and the northwestern parts of Bangladesh.

In Fig 3, the risk areas for any STH species are displayed according to the endemicity categories in Table 1. There were mainly three areas with a substantial probability (i.e., > 80%) of moderate to very high predicted prevalence. At the center of these areas are the districts of Bhola, Kishoreganj, Sunamganj, and Sirajganj. The areas in the northwestern and in the southeastern parts also showed a high probability of having a moderate or high prevalence.

## Prevalence and treatment needs by risk groups

In Table 3, the population-adjusted prevalence by risk group is summarized. To give a sense of magnitude, it was estimated that in 2020 in Bangladesh 1.6 million (95% BCI 1.4; 1.9) PSAC,

**Table 3. Population-adjusted prevalence by STH species, total number of people and number of individuals living in areas with different endemicity and annualized treatment needs by program strategy, stratified by risk group.** The reported numbers are absolute values or the median and 95%-BCI of the posterior predictive distribution.

| | | PSAC | SAC | Adults | WRA[a] | Total[b] |
|---|---|---|---|---|---|---|
| Population adjusted prevalence | *A. lumbricoides* | 9.8% (8.1%, 12.4%) | 10.6% (8.1%, 13.5%) | 9.1% (7.0%, 12.1%) | 9.5% (7.1%, 12.3%) | 9.9% (8.0%, 13.0%) |
| | *T. trichiura* | 4.4% (3.0%, 6.5%) | 5.0% (3.5%, 7.5%) | 3.7% (2.3%, 6.0%) | 3.8% (2.7%, 7.1%) | 4.3% (3.0%, 7.3%) |
| | Hookworm | 0.3% (0.2%, 0.5%) | 0.4% (0.3%, 0.6%) | 0.6% (0.5%, 0.9%) | 0.7% (0.5%, 1.1%) | 0.6% (0.4%, 0.9%) |
| | Any STH | 13.8% (11.6%, 16.4%) | 14.9% (13.0%, 18.5%) | 12.8% (10.4%, 15.2%) | 13.3% (10.7%, 17.4%) | 14.1% (11.7%, 17.6%) |
| Population (millions) | Total | 11.8 | 31.8 | 118.5 | 46.2 | 162.1 |
| | No. living in very low endemic area | 2.0 (1.1, 3.2) | 4.5 (2.1, 7.2) | 17.7 (9.8, 31.0) | 5.1 (2.2, 10.0) | 19.4 (9.7, 33.8) |
| | No. living in low endemic area | 4.9 (4.0, 5.9) | 13.3 (11.3, 15.6) | 56.6 (44.4, 66.1) | 22.4 (18.8, 25.6) | 75.3 (60.2, 88.2) |
| | No. living in moderate endemic area | 1.9 (1.6, 2.5) | 5.3 (4.1, 7.7) | 19.0 (14.6, 26.1) | 7.5 (6.0, 10.2) | 25.9 (20.3, 33.2) |
| | No. living in high endemic area | 2.2 (1.8, 2.8) | 6.3 (5.2, 8.3) | 19.4 (14.6, 25.2) | 8.4 (6.0, 10.5) | 30.8 (23.4, 40.8) |
| | No. living in very high endemic area | 0.5 (0.3, 0.9) | 1.9 (1.1, 3.1) | 4.8 (2.4, 9.3) | 2.0 (0.8, 3.9) | 7.8 (4.8, 13.9) |
| Tablets needed (millions) | | 10.6 (9.3, 12.2) | 30.5 (27.2, 36.0) | 101.6 (86.2, 116.0) | 41.6 (35.6, 48.9) | 150.6 (131.9, 178.2) |

[a]WRA are a subset of adults.

[b]The total is the union of PSAC, SAC, and adults.

4.7 million (4.1; 5.9) SAC, 15.2 million (12.3; 18.0) adults, and 6.2 million (5.0; 8.0) WRA were infected with any STH. WRA were considered a subset of adults. No statistically important prevalence differences between risk groups could be observed.

The number of people living in areas with different levels of endemicity according to Table 1 indicates that the current MDA twice per year remains appropriate for about 19.9% (95% BCI 16.3%; 26.2%) SAC. For the remaining SAC, the current treatment frequency deviates from WHO recommendations. If all SAC were reached with treatment according to WHO recommendations, it was estimated that 30.5 (95% BCI 27.2; 36.0) million tablets would be needed for MDA for SAC. Analogous reflections can be made for the other risk groups. District-level estimates of the population adjusted prevalence for any STH infection and the number of people living in very low, low, moderate, high, and very high endemic areas by risk groups are summarized in S3 Appendix.

## Discussion

This is the first study to provide species-specific estimates of STH infection prevalence at high spatial resolution across Bangladesh based on the most recent national community-based survey data of 2017–2020. Using Bayesian geostatistical modeling, we identified the areas with the highest probability to have moderate and high STH prevalence, estimated the number of infected PSAC, SAC, adults, and WRA living in moderate and high endemicity regions and estimated treatment needs for SAC at national and district levels.

Multiple geostatistical logistic regression analysis across the whole population showed that on one hand, the higher the proportion of households per district with improved sanitation, the lower the prevalence of *A. lumbricoides*. This is an established association [30] and a factor

which is taken into account by WHO's control strategy [31]. On the other hand, the higher the precipitation in the driest month of the year, the lower the prevalence of this species. We can only speculate that more precipitation in the driest month leads to a smaller chance of survival for helminth eggs outside the host or that more precipitations leads to a change in human behavior, e.g., the reduction of contacts, which might be decreasing transmission. For *T. trichiura*, the only identified risk factor was the proportion of households per district with improved sanitation, which was negatively associated with infection prevalence. As for *A. lumbricoides*, this finding is widely established in the literature [32] and was reaffirmed recently for *T. trichiura* [33]. For hookworm, the proportion of organic carbon in the soil's fine earth fraction could be identified as a statistically important risk factor, which was positively associated with the prevalence. This is in line with previous research [34] that reported a positive association with hookworm prevalence compared to *Strongyloides stercoralis* prevalence in Indonesia, but contradicts another study [35] that found a negative association for South Africa.

Comparisons of risk factors to situations in other time periods or geographic areas may be limited by different collinearity structures among the covariates [24] or different levels of treatment coverage. While a causal relationship would be reflected in a statistical association in a situation with no MDA like South Africa in 2004, the same causal relationship could be no more than random noise in a situation where the prevalence has been heavily influenced by a third factor like the consistently high national treatment coverage of MDA in Bangladesh of above 80% since 2013 [9].

The predicted geographic distribution of STH prevalence depends on the different identified risk factors and the observed prevalence for each STH species. While for *A. lumbricoides* low and high endemicity areas are scattered throughout the country, we only find two high endemicity areas for *T. trichiura* in proximity to surveyed districts and none for hookworm. These differences may be attributed to different responses to the administered anthelmintic drugs used in MDA [36], to different transmission modes or different preferred habitats for each species. The geographically uniform predicted prevalence of hookworm can also be attributed to the low variance in observed hookworm prevalence.

Previous research conducted in Southeast Asia by Lai et al. found similar geographic prevalence patterns for the respective STH species in Bangladesh [37]. However, facilitated by the availability of more and more recent survey data through the ICSPM 2017–2020, we were able to discriminate several previously unnoticed local prevalence patterns. For example, for *A. lumbricoides* and *T. trichiura*, Lai and colleagues found one broad high prevalence area covering the districts between Munshiganj and Netrakoria, while we found high prevalence around the districts of Bhola, Siranganj (in the case of *A. lumbricoides*), and the area of Kishoreganj and Sunamganj, and low prevalence in the districts between them. The detection of such local prevalence patterns allows for a more targeted allocation of treatments. Thereby the number of tablets needed for preventive chemotherapy could be reduced substantially, which is a goal put forward by the WHO STH targets for 2030 [10].

The estimation of prevalence at high spatial resolution allowed more precise calculations of treatment needs. Comparing the calculated median treatment need of 30.5 million tablets for SAC to 78.2 million actually distributed tablets in 2019 according to the WHO preventive chemotherapy database [9], shows an ongoing striking overtreatment.

While the pixel-based estimates approximate the real treatment needs best, it remains not practical, operationally. To illustrate an implementable treatment decision, the median district prevalence as reported in S4 Appendix could be used. This would imply that treatment would be stopped in two districts, while it would be continued in 30, 11, 20, and 1 districts at a frequency of every 2 years, yearly, twice a year, and thrice a year, respectively, requiring a total of

32.0 million tablets annually for SAC. Since this is less than half of the 78.2 million tablets distributed in 2019, Bangladesh would immediately achieve WHO's target of reducing the needed tablets for MDA by 50% by 2030. Furthermore, using the more conservative upper 95% BCI bound of the district prevalence estimate for a treatment decision, would imply that treatment would be continued in 16, 12, 25, and 11 districts at a frequency of every two years, yearly, twice a year, and thrice a year, respectively, requiring a total of 48.8 million tablets annually for SAC. This would still suffice for Bangladesh to immediately achieve WHO's target of reducing the needed tablets for MDA by 30% by 2025.

Economically, while assuming a price of US$ 0.25 per treatment, these two treatment decisions would reduce the annual treatment costs for SAC by US$ 11.5 million and US$ 7.4 million, respectively. Beyond the economic gains, the outlined reduction in overtreatment would also counter the growing concern of anthelmintic drug resistance. However, this result has to be taken with a grain of salt because another study, which was conducted in rural Bangladesh, found that Kato-Katz sampling leads to an overestimation of the prevalence of *A. lumbricoides* and an underestimation of the prevalence of *T. trichiura* and hookworm [38]. It is likely that our estimation data suffer from the same diagnostic uncertainties, which might lead to an overestimation or underestimation of the treatment needs. However, it is also possible, that the overestimation of the prevalence of *A. lumbricoides* and the underestimation of the prevalence of the other STH species cancel out.

One limitation of our study is that a large number of districts, especially in the northwestern and southeastern parts of the country, did not include any survey data. Predictions in those districts should be treated with caution for two reasons. First, the values of the predictors in those areas can be outside the range covered by the estimation data. For example, the eastern part of the region Chittagong includes districts with low to moderate altitudes, while the predictors at observed locations exhibit only very low altitudes. Similarly, the unobserved northwestern part and in particular the region of Rangpur feature the only districts, where open defecation is still practiced. Second, the median practical range parameter was estimated at around 100–400 km, depending on the STH species. Beyond this range, the spatial correlation becomes small, which again leads to an increased prediction uncertainty at locations far away from the observed ones. Additional surveys should be carried out in districts with sparse survey data. Moreover, a spatially more regular sampling design would allow even more precise estimates and predictions.

Another limitation is that the uncertainty in some of our covariates introduced additional uncertainty in our estimations and predictions. For example, due to the few available DHS locations in the southeastern part of Bangladesh, the district-level estimates of the WASH indicators are not robust in this area. This could be remedied by additional data collection in areas with sparse WASH data. Furthermore, the soil-related variables are predictions of a statistical model developed by ISRIC. The associated prediction error may propagate through our model and affect the prediction accuracy of our model.

## Conclusions

Our analysis of the ICSPM survey data from Bangladesh has revealed that a large amount of STH treatment among SAC is not necessary in certain districts. The estimated prevalence maps and population adjusted prevalences per district can guide treatment allocation to a more cost-effective way, reduce the administration of unnecessary treatment, and slow down potential resistance against anthelmintic drugs. Moreover, updating the control program's treatment strategy would help Bangladesh achieve the WHO's STH targets for 2030. The maps of prediction uncertainty indicate areas with sparse survey data, large variation in prevalence,

or poor fit of the models. It would be important to conduct additional surveys in those areas to better estimate the prevalence levels and therefore adjust the treatment strategy based on more solid evidence.

## Supporting information

**S1 Appendix. Covariates overview.**
(PDF)

**S2 Appendix. Bayesian geostatistical modeling.**
(PDF)

**S3 Appendix. Model estimates including risk groups.**
(PDF)

**S4 Appendix. Population adjusted prevalence and treatment needs for PSAC, SAC, adults, WRA, and any risk group by district.**
(XLSX)

## Acknowledgments

We are thankful to the health workers and field teams from the Ministry of Health & Family Welfare who conducted the surveys. The views and opinions expressed in this article are of the authors and do not necessarily reflect those of their organizational affiliations.

## Author Contributions

**Conceptualization:** Daniel J. F. Gerber, Penelope Vounatsou.

**Data curation:** Daniel J. F. Gerber, Penelope Vounatsou.

**Formal analysis:** Daniel J. F. Gerber.

**Funding acquisition:** Jürg Utzinger, Penelope Vounatsou.

**Investigation:** Daniel J. F. Gerber, Penelope Vounatsou.

**Methodology:** Daniel J. F. Gerber, Penelope Vounatsou.

**Project administration:** Penelope Vounatsou.

**Resources:** Penelope Vounatsou.

**Software:** Daniel J. F. Gerber.

**Supervision:** Jürg Utzinger, Penelope Vounatsou.

**Validation:** Daniel J. F. Gerber.

**Visualization:** Daniel J. F. Gerber.

**Writing – original draft:** Daniel J. F. Gerber.

**Writing – review & editing:** Daniel J. F. Gerber, Sanjaya Dhakal, Md. Nazmul Islam, Abdullah Al Kawsar, Md. Abul Khair, Md. Mujibur Rahman, Md. Jahirul Karim, Md. Shafiqur Rahman, M. M. Aktaruzzaman, Cara Tupps, Mariana Stephens, Paul M. Emerson, Jürg Utzinger, Penelope Vounatsou.

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
