## [Decision Letter · Decision Letter 0]

14 Apr 2023

Dear Dr. Vounatsou,

Thank you very much for submitting your manuscript "Distribution and treatment needs of soil-transmitted helminthiasis in Bangladesh: a Bayesian geostatistical analysis of 2017-2020 national survey data" for consideration at PLOS Neglected Tropical Diseases. As with all papers reviewed by the journal, your manuscript was reviewed by members of the editorial board and by several independent reviewers. In light of the reviews (below this email), we would like to invite the resubmission of a significantly-revised version that takes into account the reviewers' comments. 

We cannot make any decision about publication until we have seen the revised manuscript and your response to the reviewers' comments. Your revised manuscript is also likely to be sent to reviewers for further evaluation.

Sincerely,

Maria Victoria Periago

Academic Editor

Eva Clark

Section Editor

Your manuscript on the "Distribution and treatment needs of soil-transmitted helminthiasis in Bangladesh: a Bayesian geostatistical analysis of 2017-2020 national survey data" fits the scope of the journal and has interesting data showing the reduction in prevalence of STH´s in Bangladesh. The reviewers have made some comments and considerations that need to be addressed before a decision can be made.

Reviewer's Responses to Questions

**Key Review Criteria Required for Acceptance?**

**Methods**

-Are the objectives of the study clearly articulated with a clear testable hypothesis stated?

-Is the study design appropriate to address the stated objectives?

-Is the population clearly described and appropriate for the hypothesis being tested?

-Is the sample size sufficient to ensure adequate power to address the hypothesis being tested?

-Were correct statistical analysis used to support conclusions?

-Are there concerns about ethical or regulatory requirements being met?

Reviewer #1: Yes

Reviewer #2: 1. There is not sufficient detail about how the risk factor data was matched to Kato-Katz data with respect to space and time. Authors mention that DHS data was aggregated to district level. Were the environmental risk factors at their original level of resolution matched to individual STH data? The time period and time resolution of the risk factors is mentioned, but it is not clear whether the matching to outcome data accounted for the possibility of a long lag time between exposure and STH measurement, especially because some STH infections can last for months.

2. The unit of analysis is not clear. S2 Appendix mentions that the model was for prevalence by “survey location”. Is this district, village, or something else? This is important for understanding the purpose and appropriateness of random effects and should be mentioned at least briefly in the main text. 

3. In the methods, it would be helpful to include more detail and a citation for the Facebook population density data.

4. In Figure 2, it is not clear from the labels or caption whether these are results from bivariate models or whether models were adjusted for other covariates simultaneously. It would be preferable to adjust for other covariates in each risk factor’s model.

Reviewer #3: The methods are robust.

Line 192: Can the authors explain what a high mixing probability is and what structural zeros are?

**Results**

-Does the analysis presented match the analysis plan?

-Are the results clearly and completely presented?

-Are the figures (Tables, Images) of sufficient quality for clarity?

Reviewer #1: Yes

Reviewer #2: 1. It wasn’t clear to me why some of the variables from Fig 2 are not included in Table 2. 

2. Uncertainty estimates from the third column of Figure 1 should be discussed in more detail in the Results and Discussion section. Also, is uncertainty from the models presented in Figure 1 carried forward in the results presented in table 3? If not, this should be mentioned as a limitation.

Reviewer #3: The results are clearly presented.

**Conclusions**

-Are the conclusions supported by the data presented?

-Are the limitations of analysis clearly described?

-Do the authors discuss how these data can be helpful to advance our understanding of the topic under study?

-Is public health relevance addressed?

Reviewer #1: Yes

Reviewer #2: 1. A limitation of the study that is not mentioned is that Kato-Katz likely has low sensitivity, especially in a setting where there have been many years of MDA. Prior work in Bangladesh comparing Kato-Katz and qPCR found false negatives as well as false positives for Ascaris. At minimum, the authors should address this limitation and discuss how it could influence their findings. If the prevalence data used to fit models is under-estimated, then the estimates of necessary doses of anthelminthic treatment are likely also under-estimated. https://journals.plos.org/plosntds/article?id=10.1371/journal.pntd.0008087

2. The Discussion section on associations between sanitation and STH should also discuss the results of a randomized trial of water and sanitation and hygiene interventions in Bangladesh on STH infections and whether these align with associations in the study. https://www.ncbi.nlm.nih.gov/pmc/articles/PMC6519840/

3. In the Discussion, when talking about overtreatment, it seems important to mention increasing concerns about anthelminthic resistance.

Reviewer #3: Line 249: Please elaborate on how the parasites reproduction cycle or human behavior are expected to to explain the observed association with precipitation. What is the evidence from previous studies on the relationship between precipitation and Ascaris?

Lines 255-261: Can the authors compare and contrast the treatment coverage in Indonesia and South Africa to Bangladesh to substantiate the statement that differences in dominant risk factors may be driven by differences in treatment coverage?

**Editorial and Data Presentation Modifications?**

Reviewer #1: Yes

Reviewer #2: 1. In the Abstract “open-access, model-based databases” is vague. I suggest rephrasing and considering adding one sentence to the abstract methods on the data used.

2. In the concluding sentence of the Abstract, it should state that prevalence is estimated or model-based prevalence. 

3. Line 15: it would be helpful to clarify the ages of children treated in the national deworming program. 

4. In Appendix S1 Figures 1-2, it is not clear which time period the data was subset to or whether it was aggregated over time. 

5. Figure 1 image resolution is too low to read the legends and to see the results clearly.

6. Since DHS data was aggregated to district level, in the results, the text and figures/tables should be revised to clarify this. As written, it could imply household-level sanitation, for example.

**Summary and General Comments**

Reviewer #1: This is an interesting paper looking at a relevant, topical area. How can we measure the impact of STH programs. How can programs adjust their approach in light of new WHO treatment guidance. And how can geostatistical tools be helpful in that regard? 

Comments

• Is predicting the prevalence over a 1km x 1 km high resolution enough? STH is less focal than other diseases (such as schistosomiasis for example) of course but is greater granularity required?

• Line 18-19 – references a national treatment coverage of 98.3%. Is that for one round of MDA or a cumulative figure? That is impressively high.

• Line 29-30 and 69-70 – why were these 10 districts chosen? Were they done so with sample size calculations underpinning them, or was it from a convenience sampling approach? How comfortable are you making predictions for the South East region of the country where there are no actual samples?

• Line 33 – are these baseline figures comparing the same areas as those in the current study? 

• The estimates on the potential reduction in treatment are interesting. In the authors’ opinion is the BGM output alone enough to determine treatment approach country wide? Or do they need to be supplemented with IU/EU specific surveys required?

• Model validation, lines 116-117. ‘Model validation was performed by fitting each model to a randomly selected subset of 80% locations (training set) and predicting the disease risk at the remaining ones (testing set).’ Does that mean taking 80% of the ten districts, or 80% of the 296 villages? Or another approach?

• Lines 176 – 177. Socioeconomic factors were not significant. Isn’t that surprising?

• Lines 178-184 – this seems to repeat the information in the paragraph above. Am I confusing something?

• Table 2 – is it possible to calculate for the combined STH species? That’s what determines treatment approach.

• Lines 223-224 – are the numbers for WRA contained within the adults number or are they separate?

Minor / Editorial

Line 2 – ‘infection’ should probably be ‘infections’

Line 279 – ‘patters’ should probably be ‘patterns’

Line 292 – ‘trice’ should probably be ‘thrice’

Reviewer #2: This paper uses Bayesian geostatistical models and spatially indexed environmental data to predict STH prevalence nationwide using data from 10 out of 64 districts. This study’s findings are useful for informing whether MDA should be focally targeted and/or modified in future years. However, there are some limitations of the analysis that have not been adequately addressed. In addition, more detail is needed about the data processing to understand the risk of misclassification of the risk factors.

Reviewer #3: This is an informative and important analysis. One major limitation is that STH data are only available from 10 out of 64 districts. The authors acknowledge this in the discussion section but this information should also be included in the abstract and in the methods sections under data.

PLOS authors have the option to publish the peer review history of their article (what does this mean?). If published, this will include your full peer review and any attached files.

Reviewer #1: Yes: Michael French

Reviewer #2: No

Reviewer #3: No
---

## [Editor Report · Decision Letter 1]

11 Sep 2023

Dear Dr. Vounatsou,

We are pleased to inform you that your manuscript 'Distribution and treatment needs of soil-transmitted helminthiasis in Bangladesh: a Bayesian geostatistical analysis of 2017-2020 national survey data' has been provisionally accepted for publication in PLOS Neglected Tropical Diseases.

Best regards,

María Victoria Periago

Academic Editor

Eva Clark

Section Editor

The manuscript of this study conducted in Bangladesh to estimate the geographical distribution of STH infections using high spatial resolution, identify risk factors, and estimate treatment needs at different population subgroups. The authors have taken into consideration the comments and suggestions made by three reviewers during the first evaluation round and have addressed them fully. Therefore, I recommend acceptance.

---

## [Editor Report · Acceptance letter]

29 Oct 2023

Dear Dr. Vounatsou,

We are delighted to inform you that your manuscript, "Distribution and treatment needs of soil-transmitted helminthiasis in Bangladesh: a Bayesian geostatistical analysis of 2017-2020 national survey data," has been formally accepted for publication in PLOS Neglected Tropical Diseases.

Best regards,

Shaden Kamhawi

co-Editor-in-Chief

Paul Brindley

co-Editor-in-Chief
